Hierarchical transformer speech depression detection model research based on Dynamic window and Attention merge

Yue Xiaoping
Zhang Chunna 320073500077@ustl.edu.cn
Wang Zhijian
Yu Yang
Cong Shengqiang
Shen Yuming
Zhao Jinchi
School of Computer Science and Software Engineering, University of Science and Technology Liaoning , Anshan , Liaoning , China
Sperlì Giancarlo
Electronic publication date: 2024 Sep 26
Publication date: 2024
Volume: 10
Electronic Location ID: e2348
Received 2024 Jun 6; Accepted 2024 Aug 31
Copyright: ©2024 Yue et al.
Copyright year: 2024
Copyright holder: Yue et al.
License: This is an open access article distributed under the terms of the Creative Commons Attribution License, which permits unrestricted use, distribution, reproduction and adaptation in any medium and for any purpose provided that it is properly attributed. For attribution, the original author(s), title, publication source (PeerJ Computer Science) and either DOI or URL of the article must be cited.
License URL: https://creativecommons.org/licenses/by/4.0/

Keywords: Transformer, Speech signal processing, Speech emotion recognition, Hierarchical framework

Funding: The authors received no funding for this work.

==============================
Depression Detection of Speech is widely applied due to its ease of acquisition and imbuing with emotion. However, there exist challenges in effectively segmenting and integrating depressed speech segments. Multiple merges can also lead to blurred original information. These problems diminish the effectiveness of existing models. This article proposes a Hierarchical Transformer model for speech depression detection based on dynamic window and attention merge, abbreviated as DWAM-Former. DWAM-Former utilizes a Learnable Speech Split module (LSSM) to effectively separate the phonemes and words within an entire speech segment. Moreover, the Adaptive Attention Merge module (AAM) is introduced to generate representative feature representations for each phoneme and word in the sentence. DWAM-Former also associates the original feature information with the merged features through a Variable-Length Residual module (VL-RM), reducing feature loss caused by multiple mergers. DWAM-Former has achieved highly competitive results in the depression detection dataset DAIC-WOZ. An MF1 score of 0.788 is received in the experiment, representing a 7.5% improvement over previous research.

Introduction

According to statistics from the World Health Organization (WHO), approximately one billion people globally suffer from mental disorders. The impact of the COVID-19 pandemic has led to a surge in the number of people with mental disorders, with the number of depression patients increasing by as much as 27.6% (World Health Organization, 2022). Compared to the vast number of individuals with depression, the number of professional psychological counselors is severely inadequate. However, the diagnosis of depression relies on the subjective judgment of these counselors. This results in significant delays in diagnosis and a high probability of misdiagnosis. The need for widely deployable, low-barrier-to-use machines for the diagnosis of depression has become an urgent issue to address.

Humans express their emotions in various ways. Speech is the most direct way to express emotions, containing numerous features related to emotional expression. These features include pronunciation, intonation, and pauses. Studies have found significant differences in vocal characteristics. These differences exist between individuals with depression and healthy individuals. Therefore, speech features are considered to be one of the key objective features for analyzing depression (Shin et al., 2021; He et al., 2022). Moreover, an increasing number of studies have also attempted to employ technologies such as convolutional neural networks (CNN), long short-term memory (LSTM) networks, and others for audio-based depression detection. Researchers typically use CNNs to extract stable and effective local representations from speech features. These speech features include Mel-frequency cepstral coefficients (MFCC) and spectrograms (Othmani et al., 2021; Saidi, Othman & Saoud, 2020; Das & Naskar, 2024; Lakhan et al., 2023). This extraction helps in recognizing psychological disorders such as depression. LSTM networks, capable of capturing longer-distance dependencies, serve to compensate for the CNN’s tendency to focus excessively on local features, thereby enhancing the capability to identify depressive states (Zhao et al., 2021a).

Transformers (Vaswani et al., 2017) have been developed and applied across various domains. The utilization of global attention mechanisms, has demonstrated superior modeling capabilities. Experiments show that Transformers are better at handling long-distance dependencies within sequences compared to LSTM. This structure is depicted in Fig. 1A. However, Transformers suffer from high computational cost, especially for long features like speech signals. A sliding window attention mechanism requires less computational overhead than global attention mechanisms. This mechanism allows the model to focus on regions with key features. Due to speech has a highly unique structure, each sentence is composed of multiple words, each word consists of multiple phonemes, and each phoneme is formed by several audio frames. Considering the structural characteristics of speech, a hierarchical Transformer model has emerged for the detection of depression in speech. Following this idea, emerging speech detection models such as KS-Transformer and SpeechFormer have surfaced (Chen et al., 2022b; Chen et al., 2023; Chen et al., 2022a; Zhao et al., 2019). As illustrated in Fig. 1B, these models integrate the structural characteristics unique to speech, segmenting and merging entire speech segments into multiple stages for feature processing. However, the process of segmenting and merging original features uses sliding windows. These studies overlook the varying lengths of different words in speech signals, relying solely on fixed window sizes. These fixed window sizes are obtained indirectly through methods like P2FA (Yuan & Liberman, 2008). This approach fails to adequately segment each word, resulting in merged features that cannot effectively represent the subsequent stage of the speech signal.

Figure 1 Comparison diagram of depression detection model structures based on speech and transformer.

To address the aforementioned issue, this article integrates window attention mechanisms and reconsiders the fundamental composition of speech signals, thereby constructing a hierarchical architecture—Dynamic Window and Attention Merge Hierarchical Transformer (DWAM-Former). The encoder component of the DWAM-Former model is depicted in Fig. 1C. In DWAM-Former, the Learnable Speech Split Module (LSSM) uses textual information and timestamps to generate varying window sizes. These varying window sizes are used for each feature processing and merging stage. Then each stage of the Dynamic Window Transformer Block uses different window sizes for feature processing. Simultaneously, the Adaptive Attention Merge Module (AAM) merges original features using an attention mechanism. This merging generates more comprehensive and representative features for the subsequent stage. Additionally, the DWAM-Former model connects the original feature information, after being matched in length by the Variable-Length Residual Module (VL-RM), with the features merged at each stage. This alleviates the problem of feature loss caused by multiple segmentation and merging processes. Experimental results show that DWAM-Former achieves an MF1 score of 0.788 on the depression detection dataset DAIC-WOZ. This score represents a 7.5% improvement over previous studies.

Related Works

Since its introduction in the 1980s, deep learning has seen tremendous growth and has been widely applied in research across various fields. Neural networks form the foundation of deep learning, with CNNs being among the earliest and most extensively developed types. CNNs are feedforward neural networks that include convolutional layers and have a deep structure. The convolutional kernels in the hidden layers can learn stable local representations. This method is effective for learning and extracting speech features. The FVTC-CNN model leverages CNNs to extract full vocal tract coordination (FVTC) features from raw speech for detecting depression in noisy environments (Huang, Epps & Joachim, 2020). The EmoAudioNet model uses CNNs to learn features from raw audio spectrograms and MFCCs, then fuses these features for depression detection (Othmani et al., 2021). However, CNNs are more commonly used for image data classification tasks, whereas SVM classifiers offer better generalization capabilities and can handle a wider variety of data types. Consequently, the Saidi model uses CNNs for feature extraction from raw speech spectrograms and employs SVM classifiers for depression classification (Saidi, Othman & Saoud, 2020). Additionally, dilated convolutions introduce space dilations between kernel elements, expanding the receptive field to capture a broader range of feature information. The STFN (Han et al., 2024) model integrates causal and dilated convolutions. It continuously expands the receptive field to acquire multi-scale contextual information, showing promising detection results on the DAIC-WOZ dataset.

Speech data differs from image data because individual image regions do not have temporal characteristics. In contrast, depressed speech data often consists of longer segments, and temporal states influence each other. LSTM networks are a type of recurrent neural network. They are designed to handle tasks with long intervals and delays in time series. LSTMs can capture longer-range dependencies more effectively. The LSTMTF model utilizes openSMILE to extract frame-level audio features for input to the LSTM, combined with an attention mechanism for depression detection (Zhao et al., 2021b). Additionally, both the Solieman model (Solieman & Pustozerov, 2021) and the FCNN-LSTM model (Lakhan et al., 2023) integrate CNN and LSTM network characteristics, applying them to the detection of depression and autism in children, respectively.

Although LSTM models can capture long-range data dependencies, speech data also requires global analysis. The Transformer model, based on self-attention mechanisms, comprises encoders and decoders, each consisting of multiple layers of self-attention mechanisms and fully connected neural networks. Compared to traditional convolutional or recurrent network structures, Transformers are better equipped to handle long sequences and capture global dependencies within sequences, offering superior model expressiveness. The SIMSIAM-S model leverages Transformers to process speech features obtained through self-supervised learning methods, deriving global speech features for depression detection (Dumpala et al., 2022).

_______________________  Algorithm 1: Fixed Window Size Self-Attention Layer                   _________     input  : Q,K,V according to the feature X in length T with                  dimension dh, and the fixed window size S.     output: Feature X′ enriched by Fixed Window Size Self-Attention                  Layer.   1  for t = 1 to T do      2   Qt ∈ R1×dh ← Select the t − th token in Q; 3   Kt ∈ RS×dh ← Select (t − S 2 )th to (t + S 2 )th tokens in K; 4   Vt ∈ RS×dh ← Select (t − S 2 )th to (t + S 2 )th tokens in V ; 5   V ′t ∈ R1×dh ← Softmax(QtKT t_  √   _dh  )Vt;   6  end for  7  X′ ∈ RT×dh ← Concatenate V1′,V2′,...,V ′T;

However, audio data differs from other types of data, and the aforementioned studies have not considered its unique structural characteristics. Specifically, in the representation of a sentence in audio, a sentence is composed of multiple words, each word is made up of multiple phonemes, and each phoneme consists of multiple audio frames. Furthermore, when handling longer data, the attention range of multi-head self-attention (MSA) layers in Transformer encoders becomes very broad. This significantly increases the computational burden. As a result, researchers have explored segmenting speech data into multiple layers, leading to the development of speech detection models such as KS-Transformer and SpeechFormer (Chen et al., 2022b; Chen et al., 2023; Chen et al., 2022a; Zhao et al., 2019). In the SpeechFormer model, the self-attention layer uses a fixed window size based on MSA. This confines attention computations to a small range of adjacent tokens. As a result, the computational load is reduced. The pseudocode for the fixed window size self-attention layer is shown in Algorithm 1 . However, audio lengths vary for different words and phonemes, making it challenging to accurately fit complete words or phonemes within a fixed window size. Effective segmentation of raw speech, an efficient fusion of segmented speech features, and issues like feature detail loss after multiple layers of processing remain current research challenges. This article will address these issues to fill the gaps in existing research.

Methods

The DWAM-Former model proposed in this article, as depicted in Fig. 2, comprises three feature processing stages, two feature merge modules, and three residual connection modules. The three feature processing stages represent the processing of features for frames, phonemes, and words, respectively. In the DWAM-Former model, the dynamic window size S for all feature processing stages and merge modules is provided by the LSSM. Each specific feature processing stage is a dynamic window Transformer encoder. The calculation of multi-head attention utilizes the dynamic window size S to limit the attention calculation range. The pseudocode is similar to Algorithm 1 . Meanwhile, the feature merge modules between each feature processing stage are implemented using AAM. Additionally, in each adaptive attention merge module, as well as after the final feature processing stage, a Variable-Length Residual Module (VL-RM) has been added. This serves to mitigate the loss of feature details, thus better preserving the original feature intricacies and enhancing the final detection performance. We train and test DWAM-Former in the DAIC-WOZ dataset, which is available at https://dcapswoz.ict.usc.edu/.

Figure 2 DWAM-Former structure.

Learnable speech split module

In previous works on audio feature processing, it is often difficult to determine the length of frame features for each word. It is also challenging to know the number of phonemes in each word. Therefore, using a fixed window size for feature processing is unreasonable. To address this issue, the current study utilizes textual data corresponding to speech and additional information such as timestamps for each word, to derive the dynamic window size S for each stage of feature processing and merge modules. The structure of the learnable speech split module (LSSM) module is depicted in Fig. 3.

Figure 3 Structure of learnable speech split module.

After preprocessing the speech data, we can obtain timestamp information for each word and information about the sounds within each word. From this, we can determine the window size Sp for the phoneme stage features. Sp corresponds to the number of sounds within each word. Subsequently, the Word2Vec backbone network is employed to extract features from the sound information within each word, facilitating the transformation from text to vectors. Then, the textual features representing the phonemes, processed through the MLP network, are subjected to the Softmax function to obtain the phoneme weights. Finally, based on the weights of each phoneme within the word ωi and the frame length of the word Sw, the frame length of each phoneme Si is computed. This allows us to determine the window size Sf for frame-stage features. The specific formula for calculating the window size Sf_i of the ith feature window in the frame stage is as follows: (1) Sf_i=floorSi×ωi,i∈1,n−1Sw− ∑j=1n−1Sf_j,i=n

where Si represents the adjusted feature window of the frame stage, ωi represents the weight corresponding to the ith phoneme, and Sw represents the frame length of the word.

Adaptive attention merge

Each feature merge module facilitates better alignment between the various feature processing stages. However, if each feature merge module only uses average pooling and linear connections, it may blur feature details. This happens when merging the outputs of the previous feature processing stage. To address this issue, this article introduces the adaptive attention merge (AAM), which combines feature merge weights on top of average pooling. The structure of AAM is illustrated in Fig. 4.

Based on the dynamic window size Sf and Sw generated by the LSSM module, n features from the previous stage need to be fused into one feature for the next stage. However, this n is not fixed. Therefore, for features of different lengths (n) in different merge stages, this article designs different attention weight generation modules. The attention weight generation module comprises two fully connected layers, referred to as FC layers, to linearly transform the audio feature vectors to be merged. This module first elevates the n-dimensional features X to 2n dimensions through the first FC layer, the calculation principle of which is illustrated in Eqs. (2) and ((3). Using the same approach, the features undergo a second FC layer to reduce the dimensionality back to n dimensions. Subsequently, the features undergo mapping through a sigmoid activation function to map the linearly transformed feature vectors to the range between 0 and 1, obtaining initial feature merge weights, denoted as αi. And the final feature merge weights αi′ are obtained based on average pooling. The final feature merge weights αi′ are used to compute the weighted sum of the audio feature vector X to obtain the merged feature vector X′. The specific formulas are as shown in Eqs. (2), (3), (4), (5): (2) X∈Rn×c=X1,X2,…,Xn,Xi∈R1×c,i∈1,n

(3) Y∈R2n×c=WTX+B,W∈Rn×2n,B∈R2n×c

(4) αi′=Softmax1n+αi,i∈1,n

(5) X′∈R1×c=α1′X1+α2′X2+⋯+αn′Xn

where Y represents the features elevated to 2n dimensions, W denotes the weight matrix, and B represents the bias matrix.

Figure 4 Structure of adaptive attention merge.

Variable-length residual module

In response to the problem of differing feature lengths and the loss of original feature details resulting from the merging of features at each stage, this article proposes Variable-Length Residual Module (VL-RM). The module consists of two layers. The first layer is a variable-length average pooling layer. Since the lengths of the original features and the outputs of the merge modules between each stage are different, it is necessary to shorten the length of the original features based on the length of the features outputted by the merge module of the current stage, using a variable-length average pooling layer. The variable-length average pooling layer is depicted in Algorithm 2 . The second layer involves adding the original features processed by the first layer with the output features of the feature merging module and performing layer normalization.

_______________________________________________________________________________________________________   Algorithm 2: Variable-Length Average Pooling Layer                    _________     input  : Initial audio feature Xinitial with length linitial, and the                  feature length l’ of meraged audio feature X’.     output: Residual audio feature Xr.   1  pad_length ← 0;   2  if linitial mod l′ ⁄= 0 then      3   pad_length ← l′− (linitial mod l′); 4   Xinitial ← Add pad_length pad to the end of Xinitial;   5  end if  6  X′initial ← Group Xinitial by (linitial+pad_length           l′      );   7  Xr ← Apply AvgPool inside every groups in X′initial;

Experimental Analysis

Depression databases and evaluation metrics

The Distress Analysis Interview Corpus-Wizard of Oz (DAIC-WOZ) database (Gratch et al., 2014) is a real clinical dataset used in the Audio/Visual Emotion Challenge and Workshop (Ringeval et al., 2019) for depression recognition. The DAIC-WOZ is a subset of the Distress Analysis Interview Corpus (DAIC), renowned for its completeness, recognition, and high utilization among publicly available depression datasets. Through the interactive form of interview communication between the electronically manipulated virtual interviewer Ellie and the interviewees, audio-video data with a conversation duration exceeding 50 h was collected.The length of each audio file ranges from 7 to 33 min, with a fixed sampling frequency of 16,000 Hz. Furthermore, this dataset utilizes the Patient Health Questionnaire-8 (PHQ-8) scores from interviewees, along with a binary status label (PHQ-8 > 10 and PHQ-8 ≤ 10), as indicators of the interviewee’s mental health status. Additionally, the training and validation sets of DAIC-WOZ comprise audio data from 107 and 35 interviewees, respectively. Each speech segment in the DAIC-WOZ corpus contains the voices of both the interviewees and the interviewers, along with some irrelevant ambient noise, making it unsuitable for direct use in depression identification. Therefore, to address this issue, it is necessary to first isolate each question and answer between the interviewees and interviewers based on the timestamps in the text transcripts of the corpus, while simultaneously discarding irrelevant ambient noise. Since the DAIC-WOZ suffers from sample imbalance (100 healthy and 42 depressed interviewees) and some participants’ utterances are quite short, we resample the corpus in our experiment. In detail, when the participant is depressed, we select his/her longest 46 utterances as the training or testing samples. Otherwise, the longest 18 utterances are selected. By following the dataset split of DAIC-WOZ, we have 1,386 healthy utterances and 1,380 depressed utterances for training, while 414 healthy utterances and 552 depressed utterances for validation.

Currently, most evaluation metrics adopt accuracy, precision, recall, and F1 score. To better assess the performance of our model, and based on previous experimental experience, we have adopted three evaluation metrics: weighted accuracy (WA), unweighted accuracy (UA), and macro-average F1 score (MF1). The specific calculation are as shown in Eqs. (6), (7), (8), (9): (6) WA=1∑i=1NWi ∑i=1NWi×Accuracyi

(7) UA=1N∑i=1NAccuracyi

(8) F1=2×Precision×RecallPrecision+Recall

(9) MF1=1N∑i=1NF1i

where Wi represents the number of samples in class i, Accuracy(i) and F1(i) represent the classification accuracy and F1 score for class i, respectively.

Experimental details

Speech feature extraction

Due to the significant differences in voice features between depression patients and healthy individuals, many scholars choose to use hand-crafted (H/C) speech features for depression identification (Sun et al., 2021; Lu et al., 2022; Fan et al., 2024). Following the guidance of the extended Geneva Minimalistic Acoustic Parameter Set (eGeMAPS) (Eyben et al., 2015), many ways of processing speech signals have been explored to generate H/C features. In the specific field of depression recognition, researchers typically opt to transform speech signals into corresponding spectrograms (Spec) using the Fast Fourier Transform (FFT). However, the perceived pitch of sound by the human ear does not exhibit a linear relationship with actual sound frequency. In this case, humans are more sensitive to segments with lower frequencies and pitch. Therefore, a logarithmic operation is performed on the amplitude and frequency information obtained from the FFT. By compressing the high-frequency and high-pitched components, this logarithmic operation obtains the log Mel spectrogram (Logmel). Furthermore, researchers extract cepstral coefficients to enhance computational efficiency. They do this based on the nonlinear relationship between Mel frequencies and Hz frequencies. This results in MFCC features. Similar to MFCC features, Filter bank (Fbank) features are obtained by removing only the last step of discrete cosine transform in the MFCC feature extraction process, preserving more raw speech feature data.

However, selecting specific H/C features requires specialized medical knowledge and consumes considerable manpower. With the advancement of deep learning, pre-trained speech feature extraction methods based on self-supervised learning have become increasingly prevalent in current speech feature extraction practices, examples include Wav2Vec, Wav2Vec 2.0 (Baevski et al., 2020), and HuBERT (Hsu et al., 2021). These methods demonstrate superior detection performance compared to H/C features (Monica & Rafael, 2022; Zhu et al., 2021; Zou et al., 2022; Sharma, 2022).

This article adopts the aforementioned speech feature extraction methods for experimental comparison. Additionally, the Mel frequency bands used for extracting Logmel are set to 128. The pre-trained HuBERT model utilized for speech feature extraction is the HuBERT-large model. In this study, unless otherwise specified, the speech feature extraction method employed is the HuBERT-based approach.

Data preprocessing

The DAIC-WOZ corpus provides textual transcripts corresponding to each audio segment and timestamp information for each speech segment of the interviewee. However, it does not provide word-level timestamps, which makes it inconvenient for word-level feature processing. The article used the open-source speech recognition toolkit VOSK (VOSK, 2022) with the general English recognition model vosk-model-en-us−0.22 to transcribe each audio segment. By obtaining the timestamp information for each word, the frame feature length for each word is derived. Simultaneously, the Carnegie Mellon University Pronouncing Dictionary from the Natural Language Toolkit (Bird, Klein & Loper, 2009) is utilized to process the words recognized by VOSK, further obtaining the number of phonemes and their types for each word. This facilitates subsequent processing to derive the frame feature length for each phoneme.

Experimental parameter settings and loss functions

The DWAM-Former model proposed in this article is trained for 75 epochs using the Adam optimizer on a single RTX 3090 GPU. The batch size is set to 16, and the initial learning rate is set to 7.5 × 10−4. The DWAM-Former model utilizes eight attention heads in each of its layers. The dynamic window transformer block layer numbers N1 to N3 are {2, 2, 4} respectively for three stages.

The categorical cross-entropy loss (CCE) used in this article is one of the commonly employed loss functions for classification problems. The specific equation is shown below: (10) CCE=−1W∑w=1W ∑i=1Nywilog2ywi′

the categorical CCE measures the difference between the predicted probabilities and the true probabilities when there are W samples and N possible classes. Here, ywi′∈R1 represents the predicted probability that the w-th sample belongs to the i-th class. When i equals the true label, ywi ∈ R1 is 1; otherwise, it is 0.

Comparative experiment

To demonstrate the effectiveness of the DWAM-Former model. On one hand, this article compares the DWAM-Former model with representative models published in recent years with the DAIC-WOZ dataset. On the other hand, as different feature extraction methods can affect the model’s detection performance, this article compares various feature extraction methods to better validate the generality and scalability of the DWAM-Former model.

Compared to the widely used deep learning techniques, DWAM-Former demonstrates superior performance on the DAIC-WOZ dataset, with MF1, UA, and WA reaching 0.788, 0.876, and 0.852 respectively. Detailed experimental results comparing different models are presented in Table 1. Compared to CNN-based models (FVTC-CNN, EmoAudioNet, Saidi, and TOAT), DWAM-Former exhibits improvements in MF1, UA, and WA by 15.88%, 20%, and 15.92% respectively. Moreover, compared to Solieman, which combines CNN and LSTM models, DWAM-Former demonstrates enhancements in MF1, UA, and WA by 29.18%, 32.68%, and 29.09% respectively. In contrast, when comparing the STFN model, which combines dilated convolution and LSTM, with the DWAM-Former model, there is an improvement of 7.95% in MF1 value and 5.97% in UA. When compared to the Transformer-based SIMSIAM-S model, DWAM-Former exhibits a 21.19% improvement in WA. Furthermore, when compared to hierarchical transformer models considering speech structure (SpeechFormer and SpeechFormer++), DWAM-Former demonstrates an increase of 7.5%, 8.22%, and 10.51% in MF1, UA, and WA respectively. The above results all highlight the detection accuracy of the DWAM-Former model.

Table 1 Comparison of the latest research methods based on audio on the DAIC-WOZ dataset.

Method	Features	WA	UA	MF 1	
FVTC-CNN (Huang, Epps & Joachim, 2020)	H/C	0.735	0.656	0.640	
EmoAudioNet (Othmani et al., 2021)	H/C	0.732	0.649	0.653	
Saidi (Saidi, Othman & Saoud, 2020)	H/C	0.680	0.680	0.680	
Solieman (Solieman & Pustozerov, 2021)	H/C	0.660	0.615	0.610	
SIMSIAM-S (Dumpala et al., 2022)	HuBERT	0.703	–	–	
TOAT (Guo et al., 2022)	Wav2Vec 2.0	0.717	0.429	0.480	
SpeechFormer (Chen et al., 2022a)	H/C(Spec)	–	–	0.558	
H/C(Logmel)	–	–	0.627	
Wav2Vec	–	–	0.694	
SpeechFormer++ (Chen et al., 2023)	H/C(FBANK)	0.743	0.754	0.733	
HuBERT	0.771	0.726	0.709	
STFN (Han et al., 2024)	H/C(Logmel)	–	0.69	0.52	
Wav2Vec	–	0.71	0.55	
Wav2Vec 2.0	–	0.77	0.73	
SIDD (Zuo & Mak, 2023)	Wav2Vec	–	–	0.601	
DWAM-Former(our)	H/C(Spec)	0.613	0.609	0.592	
H/C(Logmel)	0.683	0.662	0.676	
H/C(FBANK)	0.789	0.781	0.757	
Wav2Vec	0.815	0.797	0.754	
Wav2Vec 2.0	0.844	0.796	0.769	
HuBERT	0.852	0.816	0.788	
Notes.

All models use audio as input to ensure a fair and direct comparison. H/C = HAND-CRAFTED.

Different hand-crafted speech features and pre-trained speech features based on self-supervised learning exhibit varying detection performance within the same model. The experimental data of different speech features on the DWAM-Former model and the comparison results with other models are presented in Table 1. The H/C speech features, Spec and Logmel, achieve MF1 values of 0.592 and 0.676 respectively on the DWAM-Former model. Compared to the SpeechFormer and STFN models, the MF1 values are improved by 6.09% and 7.81% respectively. Additionally, FBANK achieves an MF1 value of 0.757 on the DWAM-Former model, showing a 3.72% improvement compared to SpeechFormer++. The self-supervised pre-trained speech feature, Wav2Vec, achieves an MF1 value of 0.754 on the DWAM-Former model. Compared to SpeechFormer, STFN, and SIDD, it shows an improvement of 8.65% in MF1 value. Additionally, Wav2Vec 2.0 attains an MF1 value of 0.769 on the DWAM-Former model. Compared to TOAT and STFN, it exhibits an improvement of 5.34% in MF1 value. Furthermore, HuBERT achieves an MF1 value of 0.788 and a WA of 0.852 on the DWAM-Former model. Compared to SIMSIAM-S, WA is improved by 21.19%, and compared to SpeechFormer++, the MF1 value is enhanced by 11.14%. The aforementioned results collectively demonstrate the robust generalizability and scalability of the DWAM-Former model.

Ablation studies

Effectiveness of each proposed components

In order to ascertain the effectiveness of each module, this section conducts ablation experiments for validation. Apart from the modules under scrutiny, all other configurations of the model remain consistent. The specific experimental results are presented in Table 2.

Table 2 Comparative experimental evaluation of modules in the DWAM-former model.

LSSM	AAM	VL-RM	WA	UA	MF 1	
	baseline		0.723	0.714	0.702	
✓	–	–	0.808	0.798	0.761	
–	✓	–	0.741	0.733	0.726	
–	–	✓	0.737	0.729	0.715	
✓	✓	–	0.837	0.809	0.780	
✓	–	✓	0.822	0.801	0.773	
–	✓	✓	0.793	0.776	0.763	
✓	✓	✓	0.852	0.816	0.788	
Notes.

The baseline values for MF1, UA, and WA are derived from the SpeechFormer model (Chen et al., 2022a) using the HuBERT feature extraction method.

1. Validation of the LSSM module: In this study, the dynamic window transformer block replaces the fixed window transformer block. This allows the model to consider the structure of speech, enabling the encoding of speech features from fine to coarse through multiple layers of dynamic windowing, rather than being restricted to fixed window sizes. The experimental results indicate that compared to the baseline model, the model’s MF1 score, UA, and WA improved by 8.4%, 11.76%, and 11.76%, respectively. Compared to the models that only adopt AAM and VL-RM, the additional utilizations of LSSM increase the MF1 score by 7.4% and 8.1%. This suggests that the encoding window size has an impact on the model’s detection performance, and dynamically changing window sizes are more consistent with the characteristics of speech structure, effectively enhancing the model’s detection performance, and thus demonstrating the effectiveness of the LSSM module.

2. The effectiveness of the AAM module: Building upon the modifications made to the aforementioned modules, this article replaces the merging block in the SpeechFormer model with the Adaptive Attention merge block. This enables the merge of features between each stage to go beyond simple average pooling within a fixed window. Instead, it utilizes attention mechanisms to assign different attention weights to features within each adaptive window, building upon average pooling. Compared to the baseline, the AMM module gives a boost of 3.4% of MF1 score. Also, the experimental results demonstrate that the model’s detection performance continues to improve with the addition of the LSSM module. Specifically, the MF1 score, UA, and WA increased by 2.5%, 1.38%, and 3.59%, respectively. This indicates that the importance of multiple speech features from the upper stage to the same speech feature of the lower stage varies, thus proving the effectiveness of the AAM module in enhancing detection performance.

3. Effectiveness of the VL-RM module: Building upon the modifications made to the aforementioned modules, this article introduces a VL-RM module after each inter-stage merge block and at the end of the final stage. This enables the preservation of finer details in the speech features after undergoing multiple layers of feature processing. With only adopting the VL-RM module, the experimental results demonstrate that the module increases the MF1 value by 1.8%, compared to the baseline model. VL-RM further improves the detection performance of the model based on LSSM and AMM, with an increase of 1.03% in MF1 value, 0.87% in UA, and 1.79% in WA. This demonstrates that adding the VL-RM module effectively improves the model’s detection accuracy.

Comparison of different methods for determining window size

In this section, through ablation experiments, we validated that the method used in this article to determine dynamic window sizes is more reasonable. The specific analysis examined the performance impact of various window size determination methods on depression detection, with experimental results presented in Table 3. Where “Fixed” indicates a fixed window size determined by statistical data, “Learnable” indicates the window size estimated through a learnable approach, “Timestamp” indicates the window size determined based on word timestamps, and “Average” indicates the window size calculated as the average of word timestamps and the number of phonemes in each word.

Table 3 Comparison of different methods for determining window size S.

S f	S p	WA	UA	MF 1	
Fixed	Fixed	0.723	0.714	0.702	
Learnable	Learnable	0.731	0.755	0.716	
Average	Timestamp	0.836	0.794	0.772	
Learnable	Timestamp	0.852	0.816	0.788	

The results indicate that using a learnable method to determine window size outperforms fixed window size, with improvements of 1.99% in MF1, 5.74% in UA, and 1.11% in WA. Moreover, employing word timestamps and the number of phonemes yields better performance compared to using only the learnable method, with enhancements of 7.82% in MF1, 5.17% in UA, and 14.36% in WA. Combining these methods results in the most suitable dynamic window size that aligns with speech structure characteristics, leading to the highest model detection performance. Compared to fixed window size, this dynamic approach increases MF1 by 12.25%, UA by 14.29%, and WA by 17.84%.

Comparison of different feature merge methods

As speech features undergo multi-stage feature processing from fine to coarse, it is necessary to merge multiple features from the previous stage into one feature for the next stage. However, the importance of each feature from the previous stage for this single feature in the next stage varies. Therefore, this section compares different merge methods to validate the rationale behind the merge method used in this article. Specific experimental results are shown in Table 4.

Table 4 Effect of different merging methods on model performance.

Merging method	WA	UA	MF 1	
Average pooling	0.723	0.714	0.702	
Attention Weight-D	0.729	0.721	0.707	
Attention Weight-F	0.764	0.771	0.720	
Average Pooling + (Attention Weight–D)	0.806	0.797	0.769	
Average Pooling + (Attention Weight–F)	0.852	0.816	0.788	
Notes.

D denotes that the feature weights come from different dimensions of the feature, while F indicates that the feature weights come from different features within the same dimension.

The experimental results indicate that using only average pooling for feature merging yields the lowest performance in model detection, with MF1 values, UA, and WA being 0.702, 0.714, and 0.723, respectively. However, employing a learnable method, which assigns different attention weights to the features to be merged, results in improvements, with MF1 values, UA, and WA increasing by 2.56%, 7.98%, and 5.67%, respectively. Utilizing a learnable method for feature merging may lead the model to overly focus on certain features, thereby disregarding the importance of other features. Thus, this study combines the average pooling method with the attention weights obtained through a learnable approach to derive new feature weights. Ultimately, this results in a further improvement in model performance, with the MF1 value, UA, and WA increasing by 9.44%, 5.84%, and 11.52%, respectively.

Comparing the two different methods of obtaining feature weights, D and F, respectively determine the merge weights based on different dimensions of features and different features within the same dimension. The experimental results indicate that the feature weights obtained by method F lead to better detection performance compared to method D. Additionally, due to the variability in the number of features compared to the fixed number of feature dimensions, merging different numbers of features requires different merging networks, resulting in higher computational complexity for method F compared to method D. While method D offers a simpler network structure, its channel attention-like approach lacks robustness in merging different features compared to method F.

To further comprehend the model and ascertain the reasons behind the observed improvements, we analyze an utterance sample from DAIC-WOZ and compare the attention weights with and without AAM by visualization. The attention weights highlight the significance of each token within the model, determined by summing all the weights of the same value vector in MSA. In this utterance, there exists forced breathing which is an indicator of depression. As shown in Fig. 5, attention weights around the forced breathing become increasingly prominent with the help of the AAM module. However, without the AAM module, the higher attention weights in the first stage of the model are assimilated by the average pooling merge, stage by stage. This comparison illustrates the ability to avoid feature losses during merging operations for the AMM module.

Figure 5 Visualizations of attention weights in different stages.

Conclusion

Speech is one of the most direct ways for humans to express their emotions, and its unique structural characteristics contain a wealth of emotional information. This article proposes a hierarchical Transformer model for audio depression detection—DWAM-Former to solve the problem of inaccurately segmenting and merging. This model takes into account the natural structure of speech, utilizing three stages of feature learning from fine to coarse: frame, phoneme, and word. The dynamic segmentation module guides the window size at each stage. This improves computational efficiency, and helps with more reasonable segmentation and feature fusion across stages.

Since the influence of different features on depression detection varies and the number of features needing to be integrated from one stage to the next differs, DWAM-Former employs an adaptive self-attention fusion module to address the feature fusion issue between stages. Compared to the average pooling method, the adaptive self-attention fusion module ensures more effective feature integration between stages. Additionally, to mitigate the gradient vanishing problem, the DWAM-Former model incorporates a variable-length residual module to enhance feature learning capabilities. Experiments on the DAIC-WOZ depression dataset demonstrate that the DWAM-Former model outperforms other depression detection models, reaching 0.788 in the MF1 score.

DWAM-Former steps forward in determining window sizes adaptively in different stages. However, it still relies on an external audio-to-text module to separate audio from sentences to words. In real world cases, words in the audio may hard to be recognized, or the language of the audio may not in the pretrained data of external module. These become limitations of the DWAM-Former. In further researches, unsupervised methods like clustering could be a potential way to group feature together. The effectiveness of Hierarchical Transformer model can be enhanced by more accurate window sizes in different stages.

Additional Information and Declarations

Competing Interests

Author Contributions

Data Availability

The authors declare there are no competing interests.

Xiaoping Yue conceived and designed the experiments, performed the experiments, analyzed the data, performed the computation work, prepared figures and/or tables, authored or reviewed drafts of the article, and approved the final draft.

Chunna Zhang conceived and designed the experiments, authored or reviewed drafts of the article, and approved the final draft.

Zhijian Wang performed the experiments, performed the computation work, prepared figures and/or tables, authored or reviewed drafts of the article, and approved the final draft.

Yang Yu performed the computation work, prepared figures and/or tables, and approved the final draft.

Shengqiang Cong performed the computation work, prepared figures and/or tables, and approved the final draft.

Yuming Shen analyzed the data, authored or reviewed drafts of the article, and approved the final draft.

Jinchi Zhao analyzed the data, authored or reviewed drafts of the article, and approved the final draft.

The following information was supplied regarding data availability:

The DAIC-WOZ dataset is available at https://dcapswoz.ict.usc.edu.

The pre-processed data is available at figshare: Wang, Zhijian (2024). DWAM-Former Raw Data. figshare. Dataset. https://doi.org/10.6084/m9.figshare.26414095.v1.

The code is available at Zenodo: Zhijian Wang. (2024). EvW1998/DWAM-Former: DWAM-Former (transformer). Zenodo. https://doi.org/10.5281/zenodo.13144684.

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
