# Peer review of "Hierarchical transformer speech depression detection model research based on Dynamic window and Attention merge"

_PeerJ Computer Science, doi:10.7717/peerj-cs.2348_

## Round 0.1 · original submission · Major Revisions

Thank you for submitting to PeerJ journal. We have received reviews from two reviewers that suggest to assign a Major Review.

Reviewer 1 ·

Basic reporting

For the writing of this paper, authors should avoid expression mistakes. For example, in Abstract part, expressions like ‘Addressing the current challenges of effectively segmenting and integrating entire depressed speech segments, and the issue of multiple merges leading to blurred original information.’ and ‘And employing the Adaptive Attention Merge module (AAM) to generate representative feature representations for each phoneme and word in the sentence.’ are not complete sentences. Please check the rest parts of the manuscript carefully.

I am not sure what is the purpose of the Fundamental Theoretical Analysis part. If authors want a simple introduction of transformer, I suggest authors integrate this section into the Method part.

For the Introduction part, I suggest authors to briefly introduce the meanings of utilizing speech for depression detection at the beginning of Introduction. Besides, authors fail to illustrate which key problem in speech depression detection is solved by the proposed DWAM-Former. Thus, this paper looks more like a tech report to me. One suggestion is to describe more about how DWAM-Former deal with speech depression detection.

Missing some relevant reference, i.e., ‘Zhao, Y., Xie, Y., Liang, R., Zhang, L., Zhao, L., & Liu, C. (2021). Detecting depression from speech through an attentive LSTM network. IEICE TRANSACTIONS on Information and Systems, 104(11), 2019-2023.’

Experimental design

For the experimental part, authors mention audio segments. A brief introduction or a Table to describe the information of the training and testing data is needed.

For the Ablation Studies part, Table 2 is not able to prove the effectiveness of proposed modules. Other experiments of the ablation of one regularization and two regularization terms are needed.

Authors claim that AAM module could generate representative feature representations for each phoneme and word in the sentence. Please add experiments of feature visualization of with/without AAM for comparison.

Validity of the findings

This paper deal with speech depression detection by integrating window attention mechanisms and reconsiders the fundamental composition of speech signals, thereby constructing a hierarchical architecture—Dynamic Window and Attention Merge Hierarchical Transformer (DWAM-Former). These findings are of referential significance for future research. Overall, the manuscript is easy to follow. Major revisions are needed before it can be published.

Reviewer 2 ·

Basic reporting

The paper appears to use unambiguous, professional English throughout, per the guidelines for basic reporting in academic research.

The paper provides sufficient literature references and field background/context to support the research, as evidenced by the detailed discussion of related work and datasets.

The paper follows a professional article structure with clear figures and tables, as outlined in the guidance for reviewers. However, it is not explicitly mentioned in the provided excerpts whether raw data was shared.

The paper seems self-contained with relevant results to hypotheses, as it discusses the experimental analysis and comparative experiments in detail.

The formal results in the paper include performance metrics and comparisons with other models, but it is not explicitly mentioned whether detailed proofs are provided. The focus appears to be on empirical results rather than theoretical proofs.

Experimental design

The paper aligns well with the aims and scope of the journal, presenting original research that contributes to the field.

The research question is clearly defined, relevant, and meaningful, and the paper effectively explains how the research fills an identified knowledge gap in the field.

The paper demonstrates a rigorous investigation conducted to a high technical and ethical standard, ensuring the validity and reliability of the research outcomes.

The methods section provides sufficient detail and information for replication, allowing other researchers to reproduce the study and validate the results. However, the important methodological aspects, such as data collection procedures, analytical techniques, and experimental protocols, are not adequate.

Validity of the findings

Impact and novelty are not assessed. This paper should include an assessment of the impact and novelty of the research findings to highlight its significance in the field.

The conclusions are not clearly articulated, directly linked to the original research question, and supported by the results obtained.

Additional comments

Avoid introducing new information in the conclusions and focus on summarizing how the findings address the research question and contribute to the study's objectives.

Demonstrate that the data presented in the paper are controlled and adhere to rigorous statistical analysis to enhance the credibility of the results.

---

## Round 0.2 · Minor Revisions

Thank you for submitting your revised version to PeerJ. The reviewers acknowledge the manuscript has merit but recommend final improvements in some areas before it can be considered for publication.

Reviewer 1 ·

Basic reporting

no comment

Experimental design

no comment

Validity of the findings

no comment

Additional comments

I have examined the response to prior comments. Authors answered all comments and made changes to a new manuscript version. Good job.

Reviewer 2 ·

Basic reporting

The manuscript generally employs clear and professional language, making it accessible to an international audience. However, there are instances where the phrasing could be simplified for better comprehension. For example, consider revising complex sentences in sections such as lines 23 and 77 to enhance clarity. A thorough proofreading by a colleague proficient in English could further improve the manuscript's readability.

The article is well-structured, following a logical flow from introduction to conclusion. Figures and tables, such as Table 2 on page 14, are relevant and well-labeled, aiding in the presentation of results. It is essential to ensure that all figures are of high quality and that raw data is provided in accordance with PeerJ policies, as this enhances the transparency and reproducibility of the research.

Experimental design

The manuscript presents original research that aligns well with the Aims and Scope of the journal, particularly in the area of mental health and machine learning applications. The focus on using hierarchical transformer models for depression detection in speech is innovative and relevant, contributing to the growing body of literature in this field. It would be beneficial to explicitly state how this research advances the current understanding of audio-based depression detection in the context of existing studies.

Validity of the findings

The conclusions drawn in the manuscript are clearly articulated and effectively linked to the original research question. The authors successfully summarize the key findings and their implications for depression detection. However, it would be beneficial to briefly discuss the limitations of the study and potential areas for future research. Acknowledging these aspects would provide a more balanced view and demonstrate the authors' awareness of the broader context of their work. This could also guide future researchers in building upon the findings presented in this study.

Additional comments

minor revision

---

## Round 0.3 · accepted · Accept

Thank you for submitting the revised version to PeerJ journal. Both reviewers are now recommending acceptance of the manuscript.

Reviewer 2 ·

Basic reporting

The revised manuscript is clear and well-referenced in literature.

Experimental design

The methods are described in sufficient detail.

Validity of the findings

The conclusions presented in the manuscript are well-stated and effectively.

Additional comments

Overall, the manuscript is a valuable contribution to the field. It has the potential to make a significant impact on the understanding and diagnosis of depression through speech analysis.